# Evaluating adherence to government recommendations for post-exposure rabies vaccine among animal-bite victims: A hospital-based study in Bangladesh

Sadia Tamanna[1,2], Dilruba Yasmin[3], Sumon Ghosh[4,5], M. Mujibur Rahaman[6], Amit Kumar Dey[7], Tushar Kumar Das[4], Sukanta Chowdhury[4]*

1 Department of Pharmacy, East West University, Dhaka, Bangladesh, 2 Department of Biomedical Science, Chosun University, Gwangju, South Korea, 3 Chattogram Veterinary and Animal Sciences University, Chittagong, Bangladesh, 4 International Centre for Diarrhoeal Disease Research, Dhaka, Bangladesh, 5 Department of Public Health, The University of Tennessee, Knoxville, United States of America, 6 Directorate General of Health Services, Ministry of Health and Family Welfare, Bangladesh, 7 Department of Livestock Services, Quality Control Laboratory, Savar, Dhaka, Bangladesh

☯ These authors contributed equally to this work.
* sukanta@icddrb.org

**Data Availability Statement:** Uploaded as supplementary information file.

## Abstract

Rabies is a fatal but preventable zoonotic disease with an approximately 100% case fatality rate. The most common way to contract rabies is through the bite of a rabid animal. Post-exposure prophylaxis (PEP) by vaccination and/or immunoglobulin therapy is the most effective measure for rabies prevention. The effectiveness of vaccination depends on the level of completion of vaccination. In Bangladesh, no previous studies were conducted to evaluate adherence to government recommendations for post-exposure rabies vaccine among animal-bite cases. We conducted a cross-sectional study to collect information about adherence to government recommendations for post-exposure rabies vaccine. A total of 457 animal bite victims were selected to collect data and follow up after one month of enrollment. The majority of participants (58%, n = 265, 95% CI: 53–63%) had a history of animal bites. Most of the participants (77%) were advised to receive three doses of vaccine and 100% of them completed 3—dose of vaccine. Among the 4—dose recommended group of participants (n = 105), 78% completed full vaccination. Of the 457 participants, 20% received post-exposure vaccine on the day of bite/scratch and the majority of the participants (66%, n = 303, 95% CI: 62–71%) received post-exposure vaccine on the day between the first and third day of bite or scratch. Increasing awareness of the importance of timely vaccination is the key to reducing the time gap between animal bites and intake of the first dose post-exposure vaccine.

## Introduction

Rabies, a deadly viral zoonotic disease causes severe illness in humans and animals. It is 100% fatal in untreated cases and 100% preventable by timely vaccination [1]. Rabies is mainly transmitted to humans through the bite of infected rabid animals [2]. In spite of viable rabies

**Funding:** The authors received no specific funding for this work.

**Competing interests:** The authors have declared that no competing interests exist.

vaccines available since 1885, rabies-related mortality in humans remains a significant health concern, particularly in low-resource countries [3–6]. People become infected with rabies because of not or incorrectly administration of post-exposure prophylaxis (PEP) [7]. According to WHO, 59000 human deaths per year are recorded globally [4,8].

The incidence of rabies can be prevented or reduced by several ways such as PEP, immediate washing of the wound, mass dog vaccination, birth control of stray dogs, and improving community awareness of rabies transmission and PEP [1]. Post-exposure prophylaxis by vaccination and/or immunoglobulin therapy is the key measure for rabies prevention strategies. Globally, more than 29 million people receive rabies vaccinations every year, preventing tens of thousands of rabies-related deaths [9]. PEP is required for all category II and III exposures assessed as carrying a risk of rabies. Bangladesh currently administers the World Health Organization (WHO)-recommended two site intradermal rabies vaccine on days 0, 3, and 7 for previously unvaccinated individuals, as well as supplementary dosages of rabies immunoglobulin (RIG) for category III exposures [10]. According to the WHO, Category I, II, III are classified as- no exposure (touching, licks on intact skin), exposure (minor scratches or abrasions without bleeding), and severe exposure (single or multiple transdermal bites or scratches) respectively [11].

Bangladesh is placed third worldwide for rabies fatality, after China and India, with an annual death toll of roughly 2,100 [2]. Over 200 persons are attacked by animals each month in Bangladesh, according to sporadic hospital records, and about 2000 deaths occur from rabies per year in Bangladesh [12]. Many of them immediately take basic treatment and then go to the clinic to get vaccinations. It was reported that 14% of people died after receiving the first dosage of the anti-rabies vaccine. It was also reported that close to 69% of individuals preferred to go to traditional healers like Kabiraj and Hujur over hospitals [13]. People are reluctant and have limited knowledge of the fatality of rabies. People are less aware of the importance of full dose vaccination. No previous studies were conducted to evaluate adherence to government recommendations for post-exposure rabies vaccine among animal-bite cases. Therefore, the purpose of the present study is to examine post-exposure prophylaxis for rabies in Bangladesh. Health officials and policymakers can consider the findings of this study in developing and designing rabies prevention strategies.

## Method

### Study site and participants

We conducted this study at the Infectious Diseases Hospital (IDH), Dhaka from January to August 2022. IDH is located in the Mohakhali area of Dhaka. It was established in 1956. It's a hospital with 100 beds that provides proper treatment and care. This hospital provides very valuable information to the patients about rabies post-exposure prophylaxis according to the WHO guidelines. Patients throughout the country visit IDH, receive medical care including rabies post exposure prophylaxis as it is the largest government medical center for individuals with animal bites. Majority of the people are from Dhaka city. In this study, we only considered animal bite victims as study participants who lived in Dhaka city. We considered every patient inside Dhaka for primary screening after we obtained their consent during the study period. We enrolled all people of any age with a history of animal bites within the last 30 days. The only people living in Dhaka who visited IDH for animal bite management were selected for data collection and follow-up.

### Study design

We included patients who were given their first dose of rabies immunoglobulin (IG) or the rabies vaccine between January and August, 2022 at the research location and who only

resided in Dhaka, Bangladesh. Patients who got rabies IG at a different clinic before showing up at our study site were not included. Patients were further eliminated if they resided outside of Dhaka or if they arrived to receive a second or third dose rather than the initial dose (Fig 1). Human rabies IG and human anti-rabies vaccines were the two medications administered at the public health center's emergency department during the investigation time span.

### Data collection and statistical analysis

We used a structured questionnaire to collect information from selected animal bite victims. We collected data from every enrolled patient two times. First interview was performed on the day patients received their first dose of rabies vaccine or RIG. Follow up interview was performed after 30 days of first dose of rabies vaccine or RIG intake. The questionnaire includes several variables such as age, sex, location of present resident, phone number of patient or their family member, history of animal bite, education level, residence distance from study site, travel cost, travelling time for coming to hospital and primary animal bite management. We also recorded the animal bite management process and its results from traditional healers who used it as a secondary animal bite management as opposed than a primary bite management. Additionally, we have documented information about the vaccine and the RIG, including the number of doses and its side effects. We identified the exact reason for the delay in arriving at IDH after exposure. Costs of medicines, vaccines, lost working hours, and the costs before coming to IDH were recorded. The following categories of animal exposures were employed to classify them: bite, scratch, lick, direct contact, and unknown encounter. "Direct contact" refers to exposures where the animal made direct contact with the patient without causing any skin erosion. Additionally, we have considered the quantity of exposure, including exposure locations. For continuous variables, averages with standard deviations were used to describe the patient demographic profile, while for categorical variables, frequencies and percentages were used.

This study was approved by the Ethical Committee of East west University (IRB No. 2022-01-003-021). A written informed consent was obtained from animal bit victims after the study aims, methods, and risks were described. In the case of children under the age of 18, the assent was obtained from their parents or guardians. The consent process was assisted by a study staff member fluent in Bengali.

## Results

### Demographic characteristics

We enrolled a total of 457 animal bite victims over the study period. Of the 457 study participants, majority were male, mean age was 26 years (standard deviation ± 15.6, range: 1–70 years) and most of them (34%) were aged between 18 and 30 years. Majority of the participants were student (36%, n = 163, 95% CI: 31–40%), followed by job (22%, n = 101, 95% CI: 18–26%) and housewife (82%, n = 82, 95% CI: 15–22%). The average household income was 328 USD (range: 96–4327 USD). Average distance from participants house to hospital was 10 kilometers (standard deviation, SD: ±9.4, range: 1–94 Kilometers). The mean time required to travel from participants house to hospital was 88 minutes (SD: ±53.5, range: 10–600 minutes) (Table 1).

### History of animals' exposure and health status of animals

Majority of participants (58%, n = 265, 95% CI: 53–63%) had a history of animal bite. Almost two-thirds of participants (72%, n = 330, 95% CI: 68–76%) were affected by single bites,

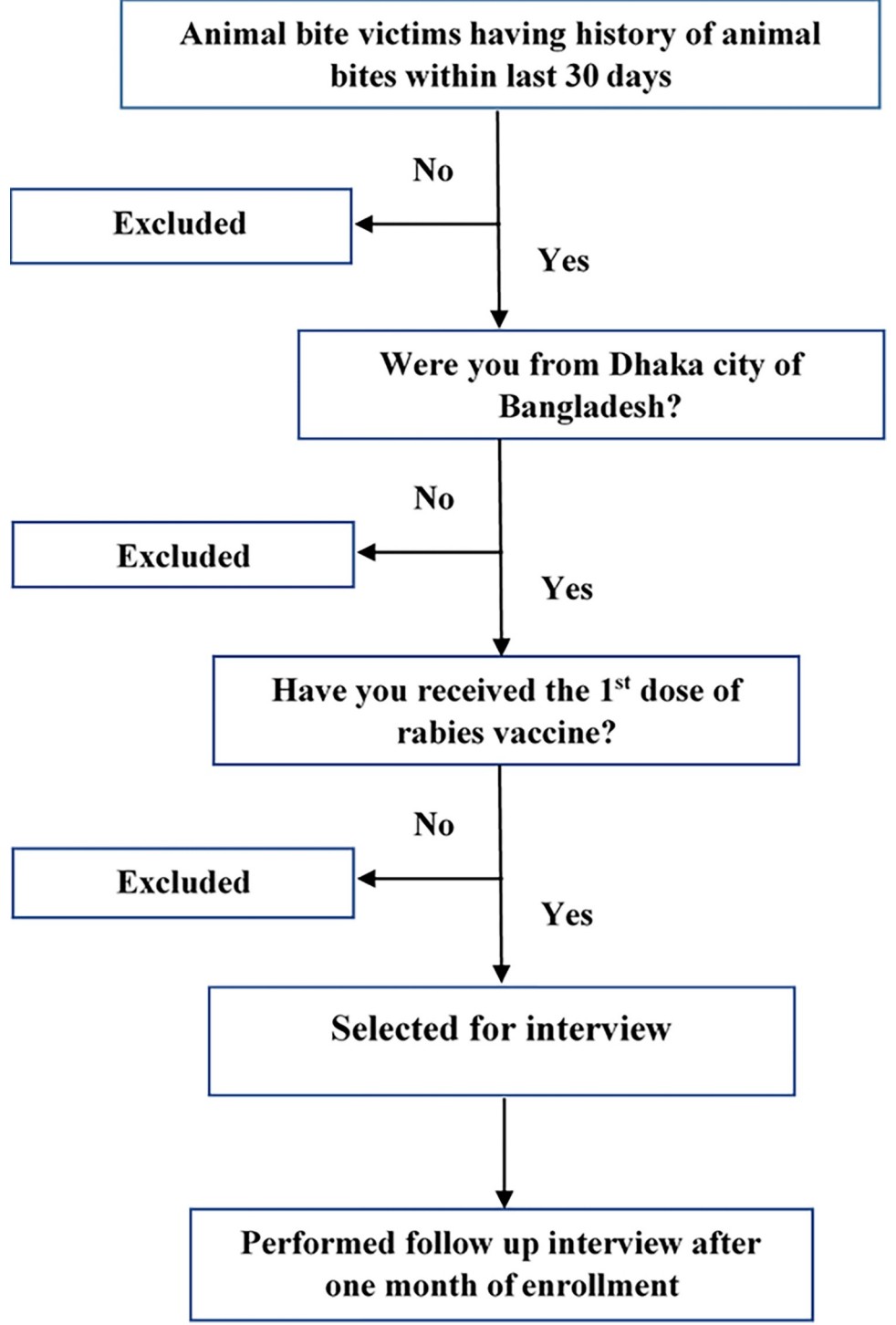

**Fig 1. Flow diagram of participant selection.**

scratches, or licks. In most cases, participants were attacked by a dog (51%, n = 233, 95% CI: 46–56%) followed by a cat (47%, n = 215, 95% CI: 42–52%). Lower (54%) and upper limbs (37%) were mostly exposed by animals. Majority of the animals were apparently healthy (85%, n = 388, 95% CI: 81–88%) during the day of bite/scratch/licking (Table 2).

**Table 1. Demographic characteristics of enrolled patients, January-August 2022 (n = 457).**

| Characteristics | Number of participants (%) | 95% CI |
|---|---|---|
| **Gender** | | |
| Male | 315 (69) | 64–73% |
| Female | 142 (31) | 27–36% |
| **Age** | | |
| 1–5 years | 34 (7) | 5–10% |
| 6–10 years | 51 (11) | 8–14% |
| 11–17 years | 68 (15) | 12–18% |
| 18–30 years | 156 (34) | 30–39% |
| 31-50Ears | 111 (24) | 20–28% |
| > 50 years | 34 (7) | 5–10% |
| **Profession** | | |
| Student | 163 (36) | 31–40% |
| Job | 101 (22) | 18–26% |
| Housewife | 82 (18) | 15–22% |
| Business | 31 (7) | 5–9% |
| Children | 30 (7) | 4–9% |
| Others | 50 (11) | 8–14% |
| **Family income** | | |
| <144 USD | 7 (2) | 1–3% |
| 145–288 USD | 266 (58) | 54–63% |
| 289–481 USD | 174 (38) | 34–43% |
| >481 USD | 10 (2) | 1–4% |
| **Education** | | |
| Illiterate | 54 (12) | 9–15% |
| Primary level | 102 (22) | 19–26% |
| Secondary level | 155 (34) | 30–38% |
| Higher secondary level | 72 (16) | 13–19% |
| Graduate level | 74 (16) | 13–20% |

## Type of wounds and wound management

Most of the participants (64%, n = 293, 95% CI: 60–69%) were classified as bites of category III, followed by category II (35%, n = 160, 95% CI: 31–40%) and category I (1%, n = 4, 95% CI: 1–2%), according to WHO. In 457 participants, 71% washed wounds with soap, 14% with only water, and 12% did nothing after immediate bite. The average time gap between bite/scratch and washing was 11 minutes (range: 1 minute– 24 hours).

## Post exposure prophylaxis compliance

Out of 457 participants, all participants received the first dose of post exposure vaccine and 389 (85%, 95% CI: 82–88%) were given rabies immunoglobulin during the day of enrollment in this study. First two months of this study period, all the victims were advised to receive 4—dose Essen Intramuscular regimen. Later, hospital authority revised the vaccine schedule from 4—dose Essen Intramuscular regimen to 3—dose Zagreb Intramuscular regimen. Over the study period, majority of the participants (n = 352, 77%) were advised to take 3—dose Zagreb regimen and 100% (n = 352) of participants completed 3—dose of vaccine. Among the 4—dose recommended group of participants (n = 105), 78% completed full vaccination.

**Table 2. Characteristics of animals' exposure and health status of animals, January-August 2022 (n = 457).**

| Characteristics | Number of participants (%) | 95% CI |
|---|---|---|
| **Types of animal exposure** | | |
| Animal bite | 265 (58) | 53–63% |
| Animal scratch | 190 (42) | 37–46% |
| Animal licking | 2 (1) | 1–2% |
| **Number of bites/scratch/lick** | | |
| Single | 330 (72) | 68–76% |
| Double | 73 (16) | 13–20% |
| Triple | 26 (6) | 4–8% |
| > triple | 28 (6) | 4–9% |
| **Site of bites/scratch/lick** | | |
| Upper limb | 171 (37) | 33–42% |
| Lower limb | 245 (54) | 49–58% |
| Head | 4 (1) | 1–2% |
| Face | 12 (3) | 1–5% |
| Neck | 2 (1) | 1–2% |
| Trunk | 13 (3) | 2–5% |
| Other site | 5 (1) | 1–3% |
| Multiple sites | 5 (1) | 1–3% |
| **Species of animals who bit participant** | | |
| Dog | 233 (51) | 46–56% |
| Cat | 215 (47) | 42–52% |
| Fox | 3 (1) | 1–2% |
| Other animals (mongoose/rat/horse/monkey) | 6 (2) | 1–3% |
| **Health status of animals during the day of attack** | | |
| Apparently healthy | 388 (85) | 81–88% |
| Showed signs of rabies | 62 (14) | 11–17% |
| Sick but not like as rabid | 7 (2) | 1–3% |
| **Reason of animal bite/scratch/licking** | | |
| Provoked by victim | 190 (42) | 37–46% |
| Provoked by animals | 65 (14) | 11–18% |
| Unprovoked | 200 (44) | 39–48% |
| Not known | 2 (1) | 1–2% |
| **Fate of attacked animals as of day of first-time data collection** | | |
| Died | 11 (2) | 1–4% |
| Killed by people | 2 (1) | 1–2% |
| Live | 224 (49) | 44–54% |
| Not found | 12 (3) | 1–5% |
| Run away | 179 (39) | 35–44% |
| Don't know | 29 (6) | 4–9% |

According to the follow-up interview, more than 90% of participants obtained vaccines on the recommended day (Table 3).

## Timeliness of post exposure vaccination and vaccination cost

Among the 457 participants, 20% received post exposure vaccine on the day of bite/scratch and the majority of the participants (66%, n = 303, 95% CI: 62–71%) received post exposure vaccine on the day between the first and third day of bite/scratch. According to the

**Table 3. Characteristics of post exposure prophylaxis compliance, January-August 2022.**

| Characteristics | Number of participants (%) | 95% CI |
|---|---|---|
| **Received first dose of rabies vaccine** | 457 (100) | - |
| **Received rabies immunoglobulin** | 389 (85) | 82–88% |
| **Recommended vaccine doses by physician** | | |
| Three doses (day 0/day 3/day 7) | 352 (77) | 73–81% |
| Four doses (day 0/day 3/day 7/day 28) | 105 (23) | 19–27% |
| **Numbers of doses received** | | |
| Received dose 1 | 457 (100) | - |
| Received dose 2 | 448 (98) | 96–99% |
| Received dose 3 | 436 (95) | 93–97% |
| Received dose 4 | 82 (18) | 15–22% |
| **Complete vaccination according to recommendation** | | |
| Received total 3 doses (n = 352) | 352 (100) | - |
| Received total 4 doses (n = 105) | 82 (78) | 69–86% |
| Did not complete (n = 105) | 23 (22) | 14–31% |
| **Vaccine received on actual recommended day** | | |
| Received 4 doses appropriately (n = 82) | 78 (95) | 88–99% |
| Received 4 doses but not appropriately (n = 82) | 4 (5) | 1–12% |
| Received 3 doses appropriately (n = 352) | 331 (94) | 91–96% |
| Received 3 doses but not appropriately (n = 352) | 21 (6) | 4–9% |

participants, 40% of the participants said that delayed vaccination was the result of other engagements (Table 4). The average cost of post exposure prophylaxis and other related costs was 5.06 USD per patient (Table 5). No cost due to loss of working time was not included during cost estimation. Vaccines and RIG for rabies at IDH were free of charge. However, each participant had to pay approximately 0.09 US Dollars for purchasing a syringe to administer vaccine and/or RIG.

**Table 4. Timeliness of post exposure vaccination and factors associated with delayed vaccine intake, January-August 2022.**

| Characteristics | Number of participants (%) | 95% CI |
|---|---|---|
| **Time gap between animal bite and first dose post exposure vaccine intake (n = 457)** | | |
| 0 day | 91 (20) | 16–24% |
| 1–3 days | 303 (66) | 62–71% |
| 4–7 days | 42 (9) | 7–12% |
| >7 days | 21 (5) | 3–7% |
| **Reasons for delayed administration of first dose of post exposure vaccine (n = 366)** | | |
| Don't know about vaccine schedule | 64 (17) | 14–22% |
| Busy with other works | 147 (40) | 35–45% |
| Took more time to find vaccine center | 10 (3) | 1–5% |
| Suffering due to animal bite | 54 (15) | 11–19% |
| Other sickness | 8 (2) | 1–4% |
| Distance to the vaccine center | 73 (20) | 16–24% |
| Government holiday | 5 (1) | 1–3% |
| Others (attending office, bad weather) | 5 (1) | 1–3% |

**Table 5. Cost associated with post exposure prophylaxis at IDH.**

| Characteristics | Average (SD) | Range |
|---|---|---|
| Travel cost from house to hospital in USD | 1.47 (2.03) | 0.09–38.46 |
| Travel cost to visit other hospital(s) before IDH visit in USD | 0.12 (0.76) | 0–9.61 |
| Financial loss because of working time loss in hour | 20 (186.89) | 0–3400 |
| Vaccine and RIG cost in USD | Free | - |
| Cost for syringe to administer vaccine and/or RIG | 0.09 (0.015) | 0–0.09 |
| Other medicine costs because of animal bite in USD | 3.31 (0.44) | 1.44–7.69 |
| Total cost associated to manage animal bite in USD | **5.06** | |

## Discussion

Prompt rabies post-exposure prophylaxis (PEP) using rabies vaccine with or without immunoglobulin is the most effective way to prevent rabies fatality. The effectiveness of PEP depends on its correct and timely administration after exposure [14]. Our study explored post-exposure compliance of rabies and determined the reason behind the delay in vaccine intake in Bangladesh. The completion rate of full dose vaccination was very high and 78–95% of animal bite victims completed. According to the PEP regime by the hospital, more than 90% of victims received vaccine appropriately. The vaccine completion rate is much higher than other low resource countries, such as Vietnam (35.6%), Iran (19%), Thailand (16%), Tanzania (28%), Haiti (54%), and Bhutan (60%) [15]. The completeness of PEP regimen was much better than Thailand where 84% completed the second scheduled dose, 70% completed the third scheduled dose, 23% completed the fourth scheduled dose, and only 16% completed the fifth scheduled dose [16]. The improvement of people's adherence to government-recommended PEP is essential to eliminate human deaths from dog-mediated rabies by 2030.

Vaccination costs may have led to poor adherence to PEP completion. Vaccines and RIG for rabies are free of charge and available at government hospitals in Bangladesh which might promote PEP completion. They need to pay very small amount of money to purchase syringe. A study from Thailand identified that high expenses for rabies PEP could be responsible for low adherence [16]. This study found that few individuals failed to receive the vaccine promptly. The results of this study reveal some specific reasons for delayed vaccination intake, such as insufficient knowledge of prompt vaccination, reluctance to receive the vaccine, and distance.

In our study, young adults (34%) and adults (24%) were most likely to become rabies victims, whereas a study reported in 2020 indicated that most victims were under 20 year old or infants in Bangladesh [13,17]. Our study also showed that most of the sufferers had completed secondary education (34%) and had a basic education. It remains significant that a large percentage of dog bites were provoked (42%). The occurrence of dog bites in young people was commonly reported among school students. In Bangladesh, rabies exposure might be prevented if victims avoided stray dogs that are in large numbers [17].

We found that the majority of participants in our study were attacked by healthy animals mostly (85%) on their lower limbs (54%). Similar output has been observed in previous literatures of rabies exposure in Bangladesh and other countries, such as Vietnam, Thailand, India [3,13,16,17]. According to a study in USA, the majority of wound locations were on the head (54%) [7]. Our analysis also determined that most of the patients (71%) mostly washed the wound area (12%) with soap, while a few patients did not do anything about the injury. In contrast, the number of people in Bangladesh who wash the bite area with soap was highly poor (2%) and most of the victims (59%) preferred to visit traditional healers instead of seeking

primary animal bite management, revealed by another previous study [17]. Surprisingly, our study found that a small number of victims visited traditional healers instead of seeking primary animal bite management.

Another critical issue in rabies post exposure prophylaxis is the timeliness of post exposure vaccination. We have found that only 20% of victims received vaccine on the day of exposure which was less than the people (66%) who receive vaccine on day 1 to 3, indicating a neglecting attitude towards rabies post exposure vaccination. Most of the victims (66%) received rabies post exposure within one to three days of exposure. A study in USA, however, showed a different outcome where 55% of victims were given vaccine on the day of animal bite exposure [7]. The failure of first dose vaccine intake on the day of exposure could be due to the attitude of ignorance and inadequate knowledge about the risk of delayed vaccination. Many victims mentioned that they did not know the actual schedule for rabies vaccination. A substantial number of participants stated that it was because of the distance from their house to the hospital. Interestingly enough, the total cost of management and vaccination was not excessive ($5.06). Getting bitten by stray and pet dogs is actually quite difficult. It is therefore recommended that all bite victims be vaccinated on the day of exposure.

This study had some limitations. Due to limited resources, we were unable to conduct this research outside of Dhaka. This Hospital study cannot compare compliance of adherence to PEP between high (bite by rabid animal) and low (bite by non-suspected rabid animal) risk groups. The results we are presenting may only represent a small number of all occurrences found in Bangladesh. There is a high level of literacy, an improved transportation system, higher income, easy access to hospitals, and an increase in awareness among people living in Dhaka that may account for the high level of vaccination compliance.

In conclusion, PEP adherence was exceptionally high. Free vaccine and a high level of awareness among victims could lead to better adherence to rabies PEP. However, a significant number of victims received the first dose of vaccine between day one and three. The time between animal bite and first dose post exposure vaccination should be reduced to zero days by improving awareness. A more in-depth understanding of potential challenges, barriers, and other associated factors is needed for PEP to be highly protective.

## Supporting information

**S1 Data. Link to datasets.**
(SAV)

## Acknowledgments

We are grateful to all study participants for providing their valuable time and information. This work was made possible by the generous support of the International Centre for Diarrhoeal Disease Research, Bangladesh.

## Author Contributions

**Conceptualization:** Sadia Tamanna, Dilruba Yasmin, M. Mujibur Rahaman, Sukanta Chowdhury.

**Data curation:** Sadia Tamanna, Dilruba Yasmin, M. Mujibur Rahaman, Sukanta Chowdhury.

**Formal analysis:** Sadia Tamanna, Dilruba Yasmin, Sumon Ghosh, Tushar Kumar Das, Sukanta Chowdhury.

**Investigation:** Sadia Tamanna, Dilruba Yasmin, Sumon Ghosh, M. Mujibur Rahaman, Amit Kumar Dey, Tushar Kumar Das, Sukanta Chowdhury.

**Methodology:** Sadia Tamanna, Dilruba Yasmin, Sumon Ghosh, M. Mujibur Rahaman, Amit Kumar Dey, Sukanta Chowdhury.

**Project administration:** Sadia Tamanna, Dilruba Yasmin, M. Mujibur Rahaman, Sukanta Chowdhury.

**Supervision:** Sukanta Chowdhury.

**Validation:** Dilruba Yasmin, Sumon Ghosh, Amit Kumar Dey, Tushar Kumar Das, Sukanta Chowdhury.

**Visualization:** Dilruba Yasmin, Amit Kumar Dey.

**Writing – original draft:** Sadia Tamanna, Tushar Kumar Das.

**Writing – review & editing:** Sumon Ghosh, Amit Kumar Dey, Sukanta Chowdhury.

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
