## [Decision Letter · Decision Letter 0]

15 May 2023

PGPH-D-23-00281

Evaluating adherence to government recommendations for post-exposure rabies vaccine among animal-bite victims: a hospital-based study in Bangladesh

Dear Dr. Chowdhury,

Thank you for submitting your manuscript to PLOS Global Public Health. After careful consideration, we feel that it has merit but does not fully meet PLOS Global Public Health’s publication criteria as it currently stands. Therefore, we invite you to submit a revised version of the manuscript that addresses the points raised during the review process.

We look forward to receiving your revised manuscript.

Kind regards,

Jianhong Zhou

Staff Editor

Journal Requirements:

1. Please provide separate figure files in .tif or .eps format only and remove any figures embedded in your manuscript file. Please also ensure all files are under our size limit of 10MB.

2. We have noticed that you have uploaded Supporting Information files, but you have not included a list of legends. Please add a full list of legends for your Supporting Information files after the references list. 

Additional Editor Comments (if provided):

Reviewers' comments:

Reviewer's Responses to Questions

**Comments to the Author**

1. Does this manuscript meet PLOS Global Public Health’s publication criteria? Is the manuscript technically sound, and do the data support the conclusions? The manuscript must describe methodologically and ethically rigorous research with conclusions that are appropriately drawn based on the data presented.

Reviewer #1: Partly

Reviewer #2: Partly

Reviewer #3: Partly

2. Has the statistical analysis been performed appropriately and rigorously?

Reviewer #1: No

Reviewer #2: I don't know

Reviewer #3: Yes

3. Have the authors made all data underlying the findings in their manuscript fully available (please refer to the Data Availability Statement at the start of the manuscript PDF file)?

Reviewer #1: Yes

Reviewer #2: No

Reviewer #3: Yes

4. Is the manuscript presented in an intelligible fashion and written in standard English?

Reviewer #1: No

Reviewer #2: Yes

Reviewer #3: Yes

5. Review Comments to the Author

Reviewer #1: The paper is an important but I have some comments and concerns

1) Consent is mentioned but ethics committee approval is not mentioned

2) The study time frame is Jan-Aug 2022. Did the pandemic have an impact on the follow up? This would become important

3) Rather than saying that the hospital follows " proper treatment" guidelines, it may be better to say " WHO guidelines for Post exposure prophylaxis were followed

4) It is not clear which regime was followed? Zagreb? Essen? IM or ID? All of this will impact adherence. 3 doses in one week or more?

5) Does the hospital give the vaccination and RIG free?

6) Multiple independent variables have been collected but I do not see a regression analysis

7) Classification of wounds into Cat I, II, III would help

8) A lot of information in there in multiple tables. Some critical information needs to be in the text too

9) Range, SD missing for many variables- for example - age, time to wound washing

Reviewer #2: This manuscript reports a prospective study that enrolled ~500 bite patients attending a busy Dhaka hospital for rabies post-exposure prophylaxis (PEP). The study reports high adherence to the WHO recommended 1-week intradermal regimen, attributed to PEP being provided for free to patients.

The study is straightforward and clear, but lacks insight that could be more valuable for improving practices and understanding the impact of health care services. I am not entirely clear on the novelty/ breakthrough reported and have some suggestions for improvements.

Based on the reported state of the biting animal, 20% were considered suspect for rabies. It would be helpful to understand what criteria were used for this classification. For example, if self-reported by the patient or probed by the clinician for symptoms diagnostic for rabies?

50% of biting animals were cats (perhaps not surprising in a setting like Dhaka), but were many cats also suspect for rabies (this is relevant because some places in Asia suggest increasing cat bites might suggest the emergence of a new reservoir)?

It would have been valuable to determine if the biting animal remained alive/healthy by day 7 (final vaccination), and thus the reliability of the health status of the animal reported (showed signs of rabies). Was this information collected?

I would be much more interested in adherence to PEP amongst this subgroup (i.e. where the animal is suspect rabid) versus all enrolled patients. Presumably this group will represent the majority of those truly at risk from rabies but protected by PEP. For those bitten by non-rabid animals, arguably poor compliance might be sensible if they know that the biting animal is healthy. It would be useful to consider risk when comparing compliance (in the first paragraph of the discussion).

The interactions amongst the different risk variables (site of exposure, provision of RIG, state of the biting animal) and delay to 1st dose are potentially insightful but appear to have not been considered.

>90% of rabies deaths are amongst bite victims that did not receive any PEP. The manuscript reports the demographics of patients who got PEP, and found them to differ from the demographics of patients who died of rabies. What does this say about health seeking from vulnerable groups? Are patients attending for PEP perhaps a somewhat different group?

Is it possible that the patient enrolment affected adherence? Would have been useful to compare with retrospective study where patients were matched for adherence.

The recommendation to reduce rabies exposures by avoiding stray dogs is naive. When a dog has rabies (including the many owned dogs that are also infected) it is actually quite hard to avoid being bitten. This may also explain the demographic of rabies deaths reported from Bangladesh!

The barriers to bite victims receiving prompt PEP also appears very superficial. I would suggest a more considered and indepth evaluation of why bite patients (specifically those at highest risk, bitten by likely rabid animals) delayed. And most importantly why rabies patients who did not receive PEP failed to do so.

Minor points

In several places I understood the authors to report PEP to comprise either vaccination or RIG. RIG without vaccine does NOT protect against rabies, so care should be taken to avoid conveying misleading information (similarly point about wound washing in 2nd paragraph of introduction).

I think it would be helpful to reflect on how practices could be further improved:

- Although RIG is only required for severe exposures, 85% of patients received RIG, but <10% had bites to the face or neck. This disparity probably needs highlighting (I didn’t see if RIG is also provided free of charge but seems like it is being overprescribed).

Cite primary studies for statistics/ reporting/ estimates; many of the references used do not relate to the point being made.

Avoid using the term ‘treatment’ for rabies. Sadly there is no treatment for this disease.

Table 5. Is the total cost for managing the animal bite excluding lost working time? I would justify and clarify this otherwise the calculation looks incorrect.

Clean up references (see for example the bad formatting for reference 9)

Data should be published with the paper!

More generally, is adherence in Dhaka likely to be higher than in other more rural parts of the country? (and is rabies incidence also likely to be lower in Dhaka given that dog vaccination is carried out (semi?) regularly in the city?

Reviewer #3: Overview and general recommendations:

The current manuscript “Evaluating adherence to government recommendations for post-exposure rabies vaccine among animal-bite victims: a hospital-based study in Bangladesh” by Tamanna S., et al. describes post-exposure prophylaxis for rabies in Bangladesh. They demonstrated that awareness is a key factor.

A similar study (Compliance to post-exposure prophylaxis among animal bite patients – A hospital-based epidemiological study) was conducted in new Delhi by Panda M and Kapoor R (J Family Med Prim Care. 2022 Oct; 11(10): 6215–6220). You can add it to your references and enrich your discussion.

After carefully analyzing the above-mentioned article, I recommend that the following points need to be addressed.

Comment 1 (general comment):

My opinion is that a better presentation and deeper analysis could improve the manuscript.

For example, 5 tables are too much. You can use figures, graphs and tables. Analysis and data interpretation could be improved.

Comment 2 (specific comments):

- Introduction line 6: “is not or incorrectly administrated” instead of is “not administered or is administered incorrectly”.

- Introduction line 6 and 7: Improve the phrase “Over 59,000 people die from rabies each year around the world” to for example “According to WHO, 59000 human deaths per year are recorded globally”.

- Introduction lines 8-10: could you please improve the phrase “The incidence of rabies can be prevented or reduced by vaccination or immunoglobulin (IG) or immediate washing the wound or animal vaccination or animal birth control and also improving awareness on rabies”

- Introduction paragraph 3: “Bangladesh is placed third worldwide” instead of Bangladesh places third worldwide”

- “Methods” instead of “Method” (titer of Methods part)

- Methods line 1: avoid repetition (improve the title)

- Results, paragraph 1, line 1: “We enrolled a total of (or in total) 457 animal bite victims” instead of “We enrolled total 457 animal bite victims”

- … I recommend further revision for English and structure

6. PLOS authors have the option to publish the peer review history of their article (what does this mean?). If published, this will include your full peer review and any attached files.

**Do you want your identity to be public for this peer review?** For information about this choice, including consent withdrawal, please see our Privacy Policy.

Reviewer #1: **Yes: **Dr Nithya Jaideep Gogtay

Reviewer #2: No

Reviewer #3: No

---

## [Decision Letter · Decision Letter 1]

27 Sep 2023

Evaluating adherence to government recommendations for post-exposure rabies vaccine among animal-bite victims: a hospital-based study in Bangladesh

PGPH-D-23-00281R1

Dear Dr. Chowdhury,

We are pleased to inform you that your manuscript 'Evaluating adherence to government recommendations for post-exposure rabies vaccine among animal-bite victims: a hospital-based study in Bangladesh' has been provisionally accepted for publication in PLOS Global Public Health.

Best regards,

Nnodimele Onuigbo Atulomah, PhD

Academic Editor

Comments from the Journal Office:

Reviewer #1 has noted that they have not been able to assess your revised version. Unfortunately the reviewer could not be contacted to assist with finding the tracked changes version of the manuscript and so this decision was made without their input.

The recommended revisions have been made

Reviewer Comments (if any, and for reference):

Reviewer's Responses to Questions

**Comments to the Author**

1. If the authors have adequately addressed your comments raised in a previous round of review and you feel that this manuscript is now acceptable for publication, you may indicate that here to bypass the “Comments to the Author” section, enter your conflict of interest statement in the “Confidential to Editor” section, and submit your "Accept" recommendation.

Reviewer #1: All comments have been addressed

Reviewer #3: All comments have been addressed

2. Does this manuscript meet PLOS Global Public Health’s publication criteria? Is the manuscript technically sound, and do the data support the conclusions? The manuscript must describe methodologically and ethically rigorous research with conclusions that are appropriately drawn based on the data presented.

Reviewer #1: Partly

Reviewer #3: Partly

3. Has the statistical analysis been performed appropriately and rigorously?

Reviewer #1: N/A

Reviewer #3: N/A

4. Have the authors made all data underlying the findings in their manuscript fully available (please refer to the Data Availability Statement at the start of the manuscript PDF file)?

Reviewer #1: Yes

Reviewer #3: Yes

5. Is the manuscript presented in an intelligible fashion and written in standard English?

Reviewer #1: Yes

Reviewer #3: Yes

6. Review Comments to the Author

Reviewer #1: The changes made are not highlighted. Also, I checked my first query on whether ethics approval was taken and this has not been mentioned in the revision. I don't know therefore if all comments have been addressed or I cannot see them because they are not highligted

Reviewer #3: Overview and general recommendations:

The current manuscript “Evaluating adherence to government recommendations for post-exposure rabies vaccine among animal-bite victims: a hospital-based study in Bangladesh” by Tamanna S., et al. describes post-exposure prophylaxis for rabies in Bangladesh. They demonstrated that awareness is a key factor.

After carefully analyzing the above-mentioned article, I recommend that the following points need to be addressed.

General comments:

- The paper was improved, you can revise it again to be more suitable for publication for example , some simple comments were not respected (such as methods and not method, in title)

- A similar study (Compliance to post-exposure prophylaxis among animal bite patients – A hospital-based epidemiological study) was conducted in new Delhi by Panda M and Kapoor R (J Family Med Prim Care. 2022 Oct; 11(10): 6215–6220). You can add it to your references and enrich your discussion.

7. PLOS authors have the option to publish the peer review history of their article (what does this mean?). If published, this will include your full peer review and any attached files.

**Do you want your identity to be public for this peer review?** For information about this choice, including consent withdrawal, please see our Privacy Policy.

Reviewer #1: No

Reviewer #3: No
